# Improved swin transformer-based thorax disease classification with optimal feature selection using chest X-ray

Nadim Rana[1]*, Yahaya Coulibaly[2]*, Ayman Noor[3], Talal H. Noor[3], Md Imran Alam[4], Zeba Khan[5], Ali Tahir[1], Mohammad Zubair Khan[6]

1 Department of Computer Science, College of Engineering and Computer Science, Jazan University, Jazan, Saudi Arabia, 2 Research and Innovation Centre, Agency of Information and Communication Technology, Bamako, Mali, 3 Department of Computer Science, College of Computer Science and Engineering, Taibah University, Madinah, Saudi Arabia, 4 Department of Electrical and Electronics Engineering, College of Engineering and Computer Science, Jazan University, Jazan, Saudi Arabia, 5 Department of Computer and Information, Applied College, Jazan University, Jazan, Saudi Arabia, 6 Department of Computer Science and Information, Applied College, Taibah University, Madinah, Saudi Arabia

* cyahaya@agetic.gouv.ml; nadimrana@jazanu.edu.sa

## Abstract

Thoracic diseases, including pneumonia, tuberculosis, lung cancer, and others, pose significant health risks and require timely and accurate diagnosis to ensure proper treatment. Thus, in this research, a model for thorax disease classification using Chest X-rays is proposed by considering deep learning model. The input is pre-processed by resizing, normalizing pixel values, and applying data augmentation to address the issue of imbalanced datasets and improve model generalization. Significant features are extracted from the images using an Enhanced Auto-Encoder (EnAE) model, which combines a stacked auto-encoder architecture with an attention module to enhance feature representation and classification accuracy. To further improve feature selection, we utilize the Chaotic Whale Optimization (ChWO) Algorithm, which optimally selects the most relevant attributes from the extracted features. Finally, the disease classification is performed using the novel Improved Swin Transformer (IMSTrans) model, which is designed to efficiently process high-dimensional medical image data and achieve superior classification performance. The proposed EnAE+ChWO+IMSTrans model for thorax disease classification was evaluated using extensive Chest X-ray datasets and the Lung Disease Dataset. The proposed method demonstrates enhanced Accuracy, Precision, Recall, F-Score, MCC and MAE of 0.964, 0.977, 0.9845, 0.964, 0.9647, and 0.184 respectively indicating the reliable and efficient solution for thorax disease classification.

**Data availability statement:** The raw image dataset used in this study is publicly available on: https://www.kaggle.com/code/anmolkamoji/lung-disease-detection/input. https://www.kaggle.com/datasets/aryashetty29/fibrosis All relevant data supporting the findings of this study are included within the manuscript.

**Funding:** The author(s) received no specific funding for this work.

**Competing interests:** The authors have declared that no competing interests exist.

## 1. Introduction

Thorax diseases encompass a broad range of conditions that affect the organs and structures within the thoracic cavity, including the lungs, heart, and surrounding tissues. Common thorax diseases include pneumonia, tuberculosis, lung cancer, pleural effusion, chronic obstructive pulmonary disease (COPD), and interstitial lung diseases [1]. These conditions can have serious health implications, ranging from breathing difficulties and chronic pain to life-threatening complications [2]. The early and accurate detection of these diseases is crucial for effective treatment and improved patient outcomes [3,4]. Traditional methods of detecting thorax diseases primarily rely on clinical assessments and imaging techniques, particularly computed tomography (CT) scans and Chest X-rays. Chest X-rays are the most commonly used imaging modality due to their accessibility, speed, and ability to reveal abnormalities in the lungs, heart, and chest wall [5]. Radiologists analyze these X-ray images for signs of disease, such as abnormal shadows, opacities, or lesions that may indicate conditions like pneumonia or lung cancer [6]. However, the manual interpretation of Chest X-rays is highly dependent on the radiologist's expertise and experience, and it can be prone to variability and errors, especially in cases of subtle or early-stage diseases [7].

Numerous research gaps exist in the classification of thoracic diseases using chest X-rays, despite advances in deep learning [8]. Because patient demographics and imaging conditions differ, models frequently do not generalize well across datasets [9]. Predictions are biased as a result of data imbalance, especially when rare diseases are underrepresented [10]. The majority of datasets only include labels at the image level, which restricts interpretability and localization [11]. Patients frequently appear with numerous illnesses, making multi-label categorization difficult for current models [12]. Deep learning systems also operate as "black boxes," providing little explanation and casting doubt on clinical reliability [13]. They could also form erroneous associations with objects that have nothing to do with illness. Practical implementation is hampered by the lack of real-world evaluation and integration into clinical operations [14]. Furthermore, the development of reliable, equitable, and thorough diagnostic tools is further limited by bias across demographic groups and the underutilization of multimodal and longitudinal data [15,16]. Building clinically dependable and morally decent AI systems for thoracic imaging requires addressing these problems.

To overcome this limitation, this research introduced a deep learning-based thorax disease classification model. The primary objective of this study is to develop a deep learning framework that is both reliable and accurate for classifying thoracic diseases from chest X-ray images. By combining an EnAE with attention processes, the study aims to enhance diagnostic performance by efficiently extracting important features that capture intricate thoracic anomalies [17]. The research employs advanced data augmentation techniques and the ChWO Algorithm to effectively select the most critical features, addressing the fundamental challenge of imbalanced datasets and enhancing the model's ability to diagnose both common and rare diseases [18,19]

accurately. Furthermore, the framework utilizes the IMSTrans model in conjunction with a multi-label classification technique to accurately classify instances that exhibit multiple concurrent states, thereby mirroring actual clinical settings. To enhance clinical applicability, the research aims to integrate explainability elements that enhance model transparency and credibility among medical professionals [20]. The major contributions of the study are:

**Design of EnAE for feature extraction**: The proposed EnAE model is designed by integrating the attention mechanism with a stacked autoencoder for extracting significant features.

**Design *of ChWO for feature selection***: The proposed ChWO algorithm is designed by incorporating the chaotic Chebyshev mapping within the existing whale optimization for obtaining a better solution in choosing the optimal best features.

**Design of IMSTrans for Thorax Disease Classification**: The proposed IMSTrans model is designed by integrating a DenseNet-based MLP layer into a conventional Swin Transformer to enhance the classification accuracy of the model.

The organization of the research is as follows: Section 2 details the related works with a problem statement, and Section 3 elaborates on the detailed Thorax disease classification. The result and discussion are presented in Section 4, and the conclusion is in Section 5.

## 2. Related works

To enhance the accuracy and efficiency of thoracic disease classification and localization from chest X-ray images, [21] designed a deep learning model named a novel attention-based convolutional neural networks (CNNs) model for thoracic disease classification (ThoraX-PriorNet). The developed model utilizes a separate network, called Anatomical Prior Estimation, which was trained to estimate disease-dependent spatial probability maps. The attention was employed to extract relevant regions in disease classification. The designed method, when applied to other medical image classification and localization tasks, is more efficient in terms of classification accuracy. The outcome relies on the precision of the estimated anatomical prior probability maps, which was the challenging aspect.

To develop an accurate and efficient deep learning model for detecting and classifying various chest diseases from X-ray images, [22] designed CXray-EffDet model using EfficientNet. In this, EfficientDet-D0 model was utilized for feature extraction based on EfficientNet-B0 to extract a distinctive set of features from the input images. It is used to predict the presence and type of chest abnormalities. The designed model offers a lightweight and computationally efficient solution for detecting and classifying chest diseases. Here, the complex structure of chest X-rays, including variations in image quality and anatomical differences, presents challenges for accurate detection and classification.

A deep learning approach based on a modified CNN with an attention mechanism based on Xception model was designed by [23] to develop a reliable and automatic system for detecting thorax disease. Here, the crucial region of the X-ray image for improved thorax detection was acquired by the attention mechanism. The model can potentially assist radiologists in chest X-ray analysis, improving efficiency and reducing workload. Datasets utilized for the evaluation have an imbalanced distribution of typical and thorax cases, requiring specific techniques to address the issue during training.

A deep learning approach based on Vision Transformers (ViTs) was designed by [24] for performing multi-label thorax disease classification tasks. In this, ViTs were pre-trained to capture the missing pixels of the input image. The designed method offers improved performance in multi-label thorax disease classification compared to traditional CNN-based approaches. The computational complexity of the model was higher when considering a large dataset.

A deep learning model based on CNN was designed by [25], wherein nine various disease classes were identified using Chest X-rays. The developed model utilized four different CNN models for extracting distinct image-level representations. The designed model demonstrates superior performance due to the consideration of various attributes. Training and deploying complex CNN models was computationally expensive, which presented a significant challenge.

A ResNet-based model was designed by [26] for thorax classification and localization using the Triplet-Attention Mechanism. The features achieved through the ResNet model were utilized to extract radiomic features from the regions highlighted by the Class Activation Maps. Finally, the classification of disease was employed through the fully connected layer.

Here, the use of radiomic features provides a more interpretable explanation of the model's predictions, helping to achieve better outcomes. Still, the computational complexity limits the model's performance.

A CX-Ultranet was designed by [27] to classify and diagnose 13 distinct thoracic illnesses from plain radiography pictures. A multiclass cross-entropy loss function was employed within a compound scaling structure, with EfficientNet serving as the baseline model. To create reduction cells and add more skip connections, channel shuffling was applied at various points along the network. Together, the loss function algorithm and Adam optimizers stabilize the model, enabling ongoing learning from new data over time.

A transfer learning method for categorizing radiological abnormalities and respiratory conditions from chest X-rays was designed by [28]. The dataset comprised 752 X-ray photos from the University Clinical Center of Kragujevac and 191,660 publicly available images. Infections, pleural thickening, atelectasis, cardiomegaly, tuberculosis, malignancies, non-viral pneumonia, and COVID-19 pneumonia, as well as healthy cases, were among the disorders it covers. This research, in contrast to others, defines up to 18 illness groups and distinguishes between healthy and diseased instances. DenseNet121 with CheXNeXt weights and additional layers for fine-tuning are used in the procedure.

CycleGAN-based preprocessing approach for enhanced lung disease classification in ChestX-Ray14 was designed by [29] that starts with identifying images with artifacts and then uses a CycleGAN model to produce sharper images in order to lessen the noise effect of the artifacts. The DenseNet-121 model, used for classification, utilizes channel and spatial attention mechanisms to focus on specific areas of the image. The model was also updated to incorporate additional data from the dataset, specifically clinical features. Table 1 shows the analysis of recent research work.

## 2.1. Problem statement

Deep learning for thoracic disease classification and localization has advanced; however, there are still a number of important research gaps that highlight the importance and applicability of further study in this area. Numerous current models, like the ResNet-based Triplet-Attention model [30] and ThoraX-PriorNet [31,32], mostly rely on anatomical priors and attention mechanisms. These methods enhance computing complexity and improve accuracy, but they also introduce

**Table 1. Analysis of Recent Research Work.**

| Authors | Methods | Key contribution | Limitations | Area under ROC (AUC) (%) |
|---|---|---|---|---|
| Hossain, M. I., *et al.* [21] | ThoraX-PriorNet | Employs attention processes and anatomical prior probability maps. | Restrictions on static priors; potential lack of generalizability across datasets | 84.67 |
| Nawaz, M., *et al.* [22] | CXray-EffDet | EfficientDet is used to classify chest diseases. | Object identification is the main focus; disease-specific interpretability is lacking. | 90.80 |
| Upasana, C., *et al.* [23] | Modified Xception Model | Attention-based categorization of pneumothorax | specifically designed for a certain illness (pneumothorax) | 90.00 |
| Xiao, J., *et al.* [24] | Masked Autoencoders | Unsupervised learning for the aspects of thoracic diseases | High processing costs and little transparency | 82.30 |
| Malik, H., & Anees, T [25] | Multi-modal DL with Cough + Imaging | Diagnoses by combining visual and auditory data. | Separate modalities are handled; fusion is not optimized. | 97.00 |
| Han, Y., *et al.* [27] | ChexRadiNet | Uses radiomics properties to classify X-rays. | Extraction of radiomics can take a long time. | 84.30 |
| Kabiraj, A., *et al.* [28] | CX-Ultranet | Multi-label detection of thoracic diseases with a small network | Accuracy may be sacrificed for efficiency; little focus | 93.00 |
| Geroski, T., *et al.* [29] | transfer learning with DenseNet121 | For chest X-rays, CNN with transfer learning | Potential underutilization of dataset-specific features | 99.00 |
| Chehade, A.H., *et al.* [30] | CycleGAN | increases image quality by augmentation for lung disease classification | increases model intricacy and the possibility of overfitting | 83.87 |

dependence on the quality of earlier predictions. Similar to this, models like CXray-EffDet [33,34] and CX-Ultranet [33,35] utilize lightweight backbones, such as EfficientNet, to promote computational efficiency; however, they still struggle to handle anatomical changes and fluctuating image quality. Although Vision Transformer-based methods provide better multilabel classification, their high computing requirements limit their scalability. Additionally, the majority of research suffers from inadequate generalizability across various clinical contexts, a lack of interpretability, and dataset imbalance. Despite recent attempts to use CycleGANs to enhance preprocessing and integrate clinical metadata, multimodal data integration is still not completely recognized. Furthermore, transfer learning techniques have the potential to benefit from massive datasets, but fine-tuning for domain-specific nuances remains a challenging task. Thus, the goal is to create a robust system that enhances diagnostic accuracy, reduces false positives and negatives, and supports radiologists in making faster and more accurate decisions, ultimately improving patient outcomes. This research is motivated by the difficulties of correctly identifying thoracic disease from chest X-ray scans, which frequently entail complicated anatomical variances and imbalanced datasets. A novel EnAE with an attention mechanism for robust feature extraction, sophisticated data augmentation techniques to reduce class imbalance, and the ChWO algorithm for optimal feature selection are all used in the proposed approach to address these problems. Ultimately, the IMSTrans model enables the accurate classification of thoracic diseases, thereby enhancing precision and facilitating more informed clinical decision-making.

## 3. Proposed methodology

The proposed thorax disease classification module comprises of four various modules like pre-processing, extraction of significant attributes, optimal feature selection and classification of thorax disease. Initially, the input Chest X-ray for processing the proposed Thorax disease classification is acquired from the dataset and is pre-processed for resizing images, normalizing pixel values, and data augmentation to improve model generalization. Thoracic diseases are rare, resulting in imbalanced datasets where certain classes are underrepresented. Then, the significant attributes are extracted using the EnAE. The EnAE is designed with a stacked auto-encoder and an attention module for enhancing classification accuracy. From the extracted features, the significant attributes are selected optimally using the ChWO algorithm. Using the extracted features, the disease classification is employed using the novel IMSTrans model. The workflow is portrayed in Fig 1.

### 3.1. Data acquisition

The input is obtained from the publicly available dataset and is utilized for the evaluation of the proposed EnAE + ChWO+IMSTrans model. The datasets like Chest X-Ray dataset [36] and Lung Disease Dataset [37] are utilized for the evaluation of the proposed EnAE + ChWO+IMSTrans-based Thorax disease classification.

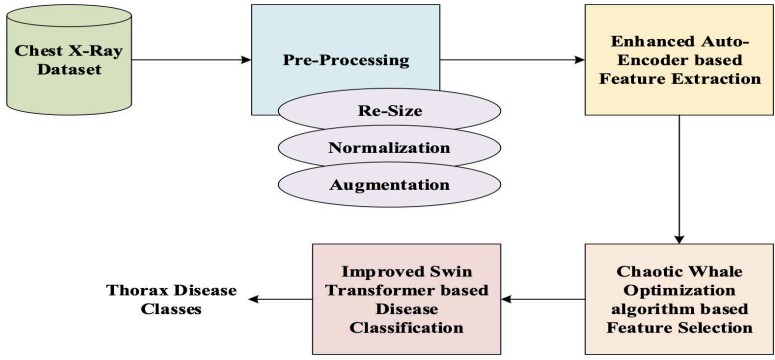

**Fig 1. Proposed Thorax Disease Detection Model.**

*ChestX-Ray dataset:* Dataset 1 contains a total of 8,274 chest X-ray images in 1024 × 1024 PNG format, covering 14 disease classes: Atelectasis, Cardiomegaly, Effusion, Infiltration, Mass, Nodule, Pneumonia, Pneumothorax, Consolidation, Edema, Emphysema, Fibrosis, Pleural Thickening, and Hernia. Each image is accompanied by rich metadata (from Data_Entry_2017.csv) including patient age, gender, view position, and more. This dataset provides a realistic and challenging basis for evaluating computer-aided diagnosis systems, due to the visual similarity of pathologies and the complexity of clinical interpretation, even when compared to CT imaging.

*Lung Disease Dataset:* This dataset was created using the NIH CXR Structured Dataset 248, which contains Chest X-ray (CXR) images designed to support the classification of lung diseases. This dataset includes a total of 4,260 high-resolution chest X-ray images, categorized into three classes: Normal Lungs, Fibrosis Lungs, and Pneumonia Lungs. The photos are organized into separate folders for each category to ensure structured access and labeling.

### 3.2. Pre-processing

Image filtering, resizing, and normalization are employed in the pre-processing stage to remove artifacts from the image and simplify further processing with minimal computational load.

**3.2.1. Gaussian filtering.** Image filtering is the process of enhancing or suppressing certain features of an image to make the relevant information more visible to the model. The proposed method utilizes Gaussian filtering to remove the artifacts from the image. The image filtered by the Gaussian filter is formulated as:

$$H'(u,v) = \frac{1}{2\pi Z^2} \exp\left(-\frac{u^2+v^2}{2 \cdot Z^2}\right) \cdot H(u,v)$$

(1)

where, the filtered outcome is signified as $H'(u,v)$, the input is denoted as $H(u,v)$, and the smoothing factor is denoted as $Z$.

**3.2.2. Resizing.** For obtaining the entire image with a unique size, image resizing is employed in the proposed disease classification model. The image is resized to 224x224 for further evaluation.

**3.2.3. Normalization.** Normalization is a process that adjusts the pixel intensity values in the image to a standard range, often between 0 and 1. The normalization is designed to make the image data more uniform and to help deep learning models converge faster and perform more effectively. The Z-score normalization is expressed as:

$$Norm(u,v) = \frac{P(u,v) - A}{SD}$$

(2)

where, the normalized outcome is denoted as $Norm(u,v)$, the average pixel intensity is defined as $A$, standard deviation is represented as $SD$ and input is signified as $P(u,v)$.

### 3.3. Data augmentation

The dataset's data is enriched to eliminate the biased outcome in the disease classification task. The proposed Thorax disease classification model employs augmentation based on rotating, flipping, cropping, shifting and scaling.

*Rotating*: Rotating of X-ray image is employed by turning the image around its center with the specified angle in a clockwise or counterclockwise direction.

*Flipping*: Flipping an X-ray image involves generating a mirror image by flipping it horizontally or vertically regardless of the orientation.

*Cropping*: Cropping an X-ray image involves cutting out a part of the image to create an augmented version that focuses on different parts and introduces variations in image framing.

 

*Shifting*: Shifting is employed by moving the X-ray image along the x-axis, y-axis, or for obtaining augmented outcome irrespective of its position within the image frame.

*Scaling*: Scaling involves enlarging or reducing the X-ray image to handle variations in the size of the regions of interest.

The augmented image and the original image in the dataset are fed into the feature extraction module to obtain the most appropriate features for reducing the computation burden of the classification model.

### 3.4. Enhanced auto-encoder-based feature extraction

The proposed EnAE model is designed by integrating the attention mechanism with the Stacked Auto-Encoders (SAEs). A feature extraction model called EnAE was created to overcome the limitations of conventional SAEs, which lose information as a result of layerwise reconstruction errors. Each EAE includes both the input from the current layer and the original raw data in the reconstruction process, in contrast to ordinary AEs that only attempt to recreate the input characteristics transmitted from earlier layers. By avoiding the cumulative degradation observed in conventional SAEs, this dual reconstruction makes sure that the learnt features at each hidden layer maintain a good representation of the original data. A deep network that can hierarchically extract progressively abstract features while maintaining the integrity of the raw input across all layers is created by stacking numerous EAEs. The significant features from the augmented and original images are extracted using the EnAE model. The structure of EnAE is portrayed in Fig 2.

*Encoder*: The encoder is responsible for transforming the input to obtain low level features and is represented through:

$$I = f(WG + d) \tag{3}$$

Here, weights and biases concerning the layer of encoder is defined as $W$ and $d$. The encoder of the EnAE model acquires $G$ as its input, and the outcome produced in the layer is signified as $I$. The output of the encoder model is obtained as a function of biases, weights, and input and is notated as $f(\cdot)$.

*Decoder*: The decoder takes the latent space representation and is represented as:

$$I = f(WI + d) \tag{4}$$

Here, the final features arrived at the outcome of the EnAE model is defined as $G'$.

*Attention Mechanism:* The attention mechanism is included after each layer of the encoder and decoder module for capturing long-range dependencies between features. Here, the inclusion of the attention layer within the stacked auto-encoder, both the attention and deep hierarchical features are extracted that assist to enhance the classification process. In addition, attention layers in both the encoder and decoder part assist to learn appropriate features in both compression and reconstruction stages of feature extraction. Here, the attention mechanism assigns different weights to each feature

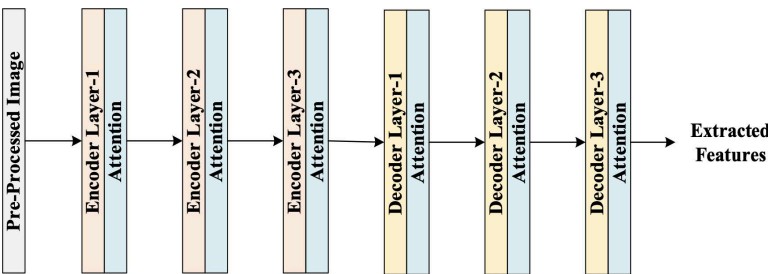

**Fig 2. Structure of EnAE model for feature extraction.**

based on its importance. Features deemed more relevant receive higher weights, while less important features receive lower weights. The attention mechanism is employed using query, key and value function. The attention score of the features is estimated as:

$$S(P, X) = P \cdot X^T \tag{5}$$

where, query and key vector is signified as $P$ and $X$ respectively and dot product is denoted as  and $T$ specifies the transformation. The Normalized form of the scores for the features is formulated as:

$$N_y = \frac{\exp\left(S\left(P, X_y\right)\right)}{\sum_x \exp\left(S\left(P, X_x\right)\right)} \tag{6}$$

where, $N_y$ is the attention weight for the $y^{th}$ input, $S(P, X_y)$ is the un-norma ized attention score for the $y^{th}$ key and the softmax function ensures that the attention weights $N$ sum to 1. Then, the outcome of the attention mechanism is defined as:

$$Attn = \sum_y N_y R_y \tag{7}$$

where, $N_y$ is the attention weight for the $y^{th}$ value and $R_y$ is the $y^{th}$ value vector. Thus, using the proposed EnAE based feature extraction model assist to capture all the significant features.

### 3.5. Chaotic whale optimization algorithm-based feature selection

The most appropriate features that enhance the Thorax disease classification are selected by the feature selection model based on ChWO algorithm [38]. The ChWO algorithm is designed by incorporating the chaotic chebyshev mapping within the existing whale optimization for obtaining a better solution in choosing the optimal features. The conventional whale optimization [39,40] is designed based on the foraging behavior of humpback whales. Humpback whales generate distinctive bubbles in the 9-shaped or circular path to capture the target. The hunting method devised by the whales termed the bubble-net feeding strategy is utilized to solve to address the optimization issue. Here, the inclusion of Chebyshev-based randomization enables the algorithm to explore a larger search space and addresses the issue of local optimal trapping.

ChWO improves the randomization criteria by introducing chaotic Chebyshev mapping into the traditional whale optimization process. This lessens the possibility of being stuck in local optima, a typical drawback of many optimization strategies, and enables the algorithm to more efficiently traverse a larger search space. The whale optimization process's rate of convergence is accelerated by the Chebyshev mapping. The ChWO ensures effective optimization during feature selection since it converges more quickly than the conventional whale optimization algorithm. By selecting the most relevant and ideal features from the extracted set, ChWO feature selection greatly enhances the classification of thoracic diseases.

The ChWO algorithm considers the current best solution as the prey and the other members of the candidates update the solution using the following equations:

$$S = \left| R \cdot Q^*(p) - Q(p) \right| \tag{8}$$

$$Q(p+1) = Q^*(p) - M \cdot S \tag{9}$$

Here, $Q^*(p)$ is the good solution obtained, $p$ denotes the present iteration, $Q(p)$ signifies the solution vector, and $M$ and $S$ signifies the coefficient vectors and is formulated as:

$$M = 2m \cdot g - m \tag{10}$$

$$R = 2g \tag{11}$$

where, $m$ decreases linearly, and $g$ is an arbitrary factor within [0,1].

*Randomization*: The randomization phase of the ChWO algorithm considers two various strategies in acquiring the global best solution. Reducing the value of $m$ to shrink the encircling range is employed in its first strategy. Then, spiral movement-based position updation is devised in its second strategy. Calculating the distance $S'$ between the candidate and target and updating the solution in a spiral-shaped path:

$$Q(p+1) = S' \cdot e^{kq} \cdot \cos(2\pi q) + Q^*(p) \tag{12}$$

where, $S' = |Q^*(p) - Q(p)|$, $k$ is the spiral movement, and $q$ is a arbitrary factor in [−1, 1]. Here, the chebyshev mapping for enhancing the randomization criteria of the whale optimization is expressed as:

$$Q(p+1) = \cos(X \cdot \cos^{-1} Q(p)) \tag{13}$$

where, the control parameter for enhancing the convergence rate is defined as $X$. Then, the solution accomplished by the proposed ChWO algorithm is signified as:

$$Q(p+1) = 0.5[Q(p+1)]_{chebyshev} + 0.5[Q(p+1)]_{Whale} \tag{14}$$

$$Q(p+1) = 0.5\left[\cos(X \cdot \cos^{-1} Q(p))\right]_{chebyshev} + 0.5\left[S' \cdot e^{kq} \cdot \cos(2\pi q) + Q^*(p)\right]_{Whale} \tag{15}$$

Thus, after the addition of chebyshev mapping, the proposed ChWO algorithm enhances the randomization criteria in capturing the global best solution.

**Local Search**: Whales search for prey randomly based on their positions. This is represented as:

$$Q(p+1) = Q_r - M \cdot S \tag{16}$$

where, $Q_r$ is a random solution.

*Termination*: The global best solution accomplishment or completion of pre-defined iteration terminates the processing. The pseudo-code is presented in Algorithm 1.

**Algorithm 1.** Pseudo-code of ChWO algorithm

```
1 Initialize the parameters, population and iteration
2 Locate the whales in the search area
3 Evaluate the feasibility based on error
4 While p < pmax
5 {
6 Estimate the solution in randomization phase using Q(p+1) = 0.5[cos(X · cos⁻¹Q(p))]chebyshev + 0.5[S' · e^kq · cos(2πq) + Q*(p)]Whale
7 Estimate the solution in local search phase using Q(p+1) = Qr − M · S
```

```
8 Re-estimate the feasibility using error
9 Return the best solution
10 }
11  p = p + +
12 end
```

Thus, using the solution accomplished by the ChWO algorithm, the optimal best features are selected for further processing.

### 3.6. Improved swin transformer-based neural network for thorax disease detection

Using the selected features, the Thorax disease detection is devised using the proposed IMSTrans model. The proposed IMSTrans is more efficient in image-based disease classification tasks because it extracts the features of the images hierarchically by capturing fine details and abstract features at multiple scales that assist in enhancing the disease classification task more efficiently. The designed model utilizes two various attention mechanisms for capturing both local and global features. In this, the non-overlapping windows-based local self-attention is utilized for capturing the local features. In addition, the shifted window mechanism with cross-window attention is employed for capturing the long-term dependent features. In addition, the extraction of complex features from the input are extracted by incorporating DenseNet within the standard Swin transformer. The DenseNet-based model assist in capturing the significant attributes with minimal computation burden through the skip connections and it assist in solving the issue concerning the vanishing gradient problem. The structure of the proposed IMSTrans model for the Thorax disease classification is portrayed in Fig 3.

The key components of the proposed IMSTrans model for Thorax disease classification are defined as:

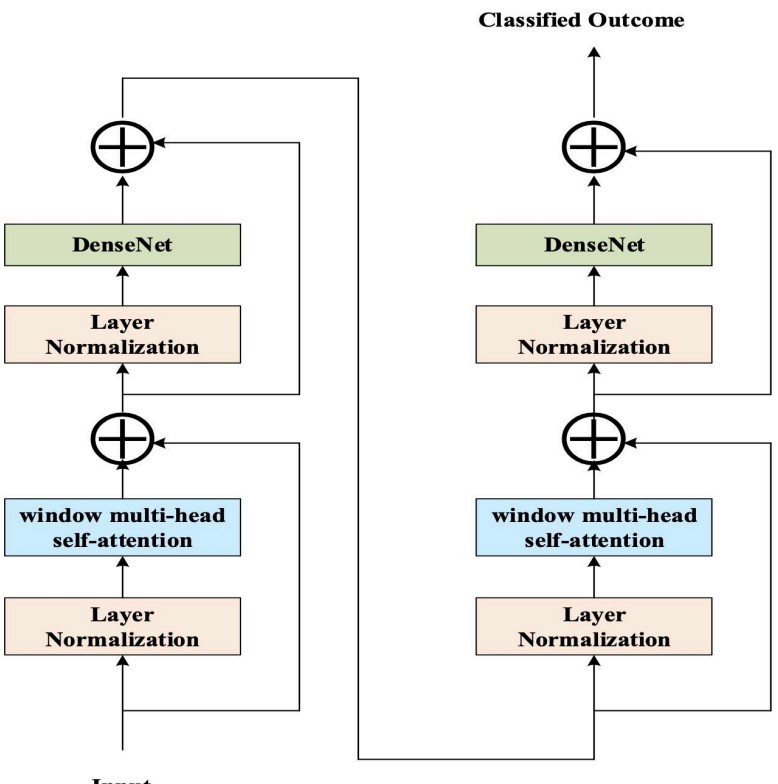

**Fig 3. Structure of proposed IMSTrans model for Thorax disease classification.**

*(i) DenseNet*: DenseNet model with its dense connectivity and efficient parameter usage is helpful in classifying the Thorax disease by extracting the complex features. In addition, the issues concerning the over-fitting is solved by the DenseNet through the efficient use of parameters helps reduce the risk of over-fitting. Also, the improved gradient flow of the DenseNet due to dense connections ensures more effective training that helps in alleviating the vanishing gradient problem. The feature re-use capability of the DenseNet makes the proposed model to learn complex features and design of DenseNet is portrayed in Fig 4.

*Dense Block*: The dense block associated with the DenseNet comprises of several convolution layers with ReLU activation and batch normalization functions. The Dense block concerning the DenseNet results in improved gradient flow during training and encourages feature reuse. The structure of dense block is portrayed in Fig 5.

*(a) Convolution Layer*: A convolution layer applies convolution operations to extract feature maps from the input. It performs a dot product between the filter (kernel) and local regions of the input image. The convolution layer in the proposed Thorax disease classification model assists in extracting the significant features concerning the texture and patterns of input features.

$$O(m,n) = \sum_{x=-l}^{l} \sum_{y=-l}^{l} F(m+x, n+y) * K(x,y)$$

(17)

where, the kernel is defined as $K(x,y)$, the input feature is defined as $F(m+x, n+y)$, and the outcome of the convolutional layer is notated as $O(m,n)$. The spatial coordinates are represented as $m$ and $n$, respectively.

*(b) Batch Normalization*: Batch normalization normalizes the output of a previous activation layer by maintaining the output 0 mean and standard deviation of 1. It is commonly applied before the activation functions in the network.

*(c)ReLU Activation*: ReLU is applied to the feature maps to ensure non-linearity.

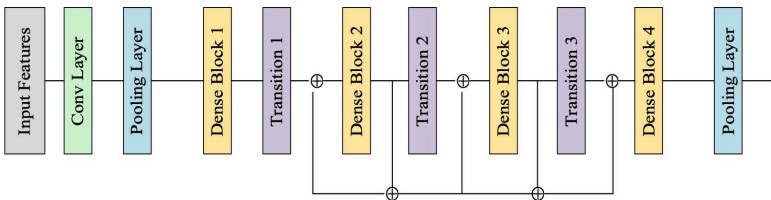

**Fig 4. Design of DenseNet.**

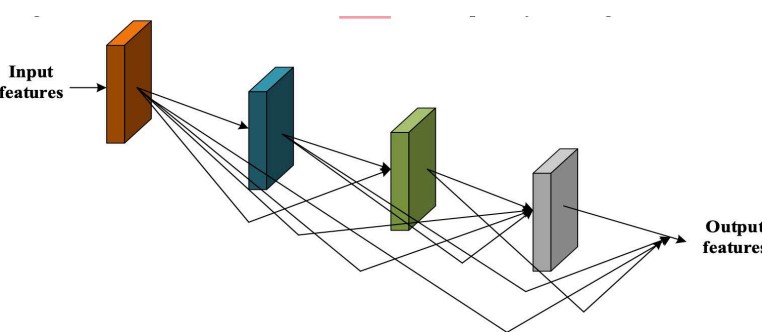

**Fig 5. Structure of Dense Block.**

*Transition* Layer: After each dense block, the transition layer reduces the dimensions through down-sampling. The transition layer helps in handling large inputs and ensures that the model remains computationally efficient.

*Pooling Layer:* Global Average Pooling (GAP) is utilized in DenseNet to reduce the feature map by averaging all elements within each feature map, resulting in a single feature per map. In the proposed model, GAP is used towards the end of the network is used for drastically reducing the number of parameters while retaining important information. In thorax disease classification, GAP helps by reducing the risk of over-fitting while still preserving the significant information necessary for accurate classification.

**(ii) Attention Mechanism based on the shifted window**: Unlike the traditional attention mechanism employed in the transformers, swin transformers utilize an attention mechanism based on the shifted window. It performs the attention mechanism within small local windows. In this, the input feature is divided into $P \times P$ non-overlapping blocks and then, it is shifted into half of the window size and is represented as $\left(\frac{P}{2} \times \frac{P}{2}\right)$. The shift in window position causes the windows to overlap, introducing connections between patches from neighboring windows. It is formulated as:

$$\hat{K}_r = W\left(L\left(\hat{K}_{r-1}\right)\right) + \hat{K}_{r-1}$$

(18)

$$K_r = D\left(L\left(\hat{K}_r\right)\right) + \hat{K}_r$$

(19)

$$\hat{K}_{r+1} = S\left(L\left(\hat{K}_r\right)\right) + K_r$$

(20)

$$\hat{K}_{r+1} = D\left(L\left(\hat{K}_{r+1}\right)\right) + \hat{K}_{r+1}$$

(21)

where, $K_{r-1}$, $K_r$, and $K_{r+1}$ represent the feature outputs at different layers and $\hat{K}_r$ represent the output of the attention module. $W$ refers to the regular window-based self-attention, and $S$ refers to the self-attention based on shifted window. The dense representation of the MLP layer is signified as $D$.

**(iii) Layer Normalization:** Layer normalization helps to standardize the input features across different image patches, ensuring that the attention mechanism can focus on meaningful relationships between patches.

**(iii) Fully Connected Layer**: In the Thorax disease classification tasks, the fully connected layer serves as the output layer. For performing the thorax disease classification task, the fully connected layer provides output through the probability distribution across various disease classes.

## 4. Result and discussion

The implementation of the proposed EnAE + ChWO+IMSTrans-based Thorax disease classification is employed in PYTHON programming language. The analysis is devised for various training and testing percentages and the outcome derived by the testing is presented in this section. The proposed method is compared with existing Thorax disease detection methods like ThoraX-PriorNet, CXray-EffDet, Attention-based CNN, and Swin Transformer to portray the superiority of the EnAE + ChWO+IMSTrans-based Thorax disease classification model.

### 4.1. Dataset description

The datasets, such as the ChestX-Ray dataset and the Lung Disease Dataset, are utilized for the evaluation of the proposed EnAE + ChWO+IMSTrans-based Thorax disease classification.

#### 4.1.1. Original data distribution. Dataset 1: ChestX-Ray dataset

- The dataset consists of 8,274 chest X-ray images across 14 thoracic disease categories:

- Atelectasis, Cardiomegaly, Effusion, Infiltration, Mass, Nodule, Pneumonia, Pneumothorax, Consolidation, Edema, Emphysema, Fibrosis, Pleural Thickening, and Hernia.

- These images are annotated with one or more disease labels, and the dataset is inherently multi-label and imbalanced. For instance, diseases such as Infiltration and Effusion are more frequent, while Hernia and Fibrosis are rare.

- This imbalance was initially visualized through class-wise frequency plots to better understand the skewness in data distribetter.

#### Dataset 2: Lung Disease Dataset

- This dataset contains 4,260 high-resolution chest X-ray images organized into three classes:

  - Normal Lungs (no abnormalities),

  - Fibrosis Lungs and

  - Pneumonia Lungs

- The data also exhibits imbalance, with Normal cases being more prevalent compared to Fibrosis and Pneumonia.

#### 4.1.2. Data distributions and preprocessing steps. Dataset 1: ChestX-Ray dataset

- **Source**: Publicly available NIH ChestX-ray14 dataset.

- **Subset Size Used**: 8,274 chest X-ray images.

- **Classes** (14 in total):

  Atelectasis, Cardiomegaly, Effusion, Infiltration, Mass, Nodule, Pneumonia, Pneumothorax, Consolidation, Edema, Emphysema, Fibrosis, Pleural Thickening, Hernia.

- **Nature**: Multi-label, class-imbalanced dataset.

  Infiltration and Effusion are overrepresented, while Hernia and Fibrosis are rare.

- **Label Handling**: The "Finding Labels" column is parsed to assign binary multi-label vectors for each class.

- **Preprocessing:**

  - Images with missing or unreadable files were removed.

  - Images resized to 256×256, followed by random crop (224×224).

  - Pixel normalization with ImageNet statistics.

  - Missing or null labels were excluded.

  - **Data Augmentation**:

  - Random horizontal and vertical flips (p=0.5).

  - Random rotations (up to ±30 degrees).

  - Augmentation increases feature variability but does not alter label distribution.

Dataset 2: Lung Disease Dataset

- **Source**: dataset containing **4,260 images**.

- **Classes**: Normal, Pneumonia, and Fibrosis.

- **Nature**: Multi-label, class-imbalanced dataset.

- **Preprocessing:**

  •Images resized and normalized using the same procedure as Dataset 1 to ensure consistency.

  •Verified for label completeness and removed any ambiguous or mislabeled samples.

**4.1.3. Data augmentation and impact.** To address the data imbalance and enhance model generalization, a robust augmentation pipeline was applied during training that includes:

- Spatial transformations (resizing, random cropping),

- Geometric variations (horizontal/vertical flipping, random rotations),

- Normalization to standardize image intensity.

These augmentations are applied dynamically at each training epoch. While the underlying image count remains unchanged, the model is exposed to diverse representations of each sample, especially from minority classes. This synthetically balances the training data, increases intra-class variance, and reduces overfitting.

**4.1.4. Hyperparameter tuning and cross-validation.** Enhancing the input feature space through feature selection based on the ChWO algorithm was the main objective of the optimization strategy. By ensuring that only the most relevant features were included for classification, this technique successfully reduced feature redundancy, enhanced class separability, and improved overall model performance.

- **Batch Size and Learning Rate:** To balance memory limitations with training stability, we chose widely used empirically validated settings a fixed learning rate of 1e-4 and a batch size of 32.

- **Optimizer:** The Adam optimizer was chosen because of its demonstrated convergence efficiency in deep learning problems, and it has a weight decay of 1e-5.

- **Training Epochs:** Up to 100 training epochs were used to give the models enough time to converge without being overfit.

## 4.2. Comparative assessment

The comparative assessment is devised using two various datasets like D1 and D2 with the existing Thorax disease detection methods like ThoraX-PriorNet [21], CXray-EffDet [22], Attention-based CNN [23], and Swin Transformer [26]. The definition for the assessment measures is portrayed in Table 2.

**4.2.1. Assessment using dataset 1.** The assessment of Thorax disease classification method using the D1 dataset is evaluated based on training data and is elaborated in this section.

The accuracy-based performance evaluation of several models at varying training percentages is displayed in Fig 6. Accuracy defines the proportion of correct predictions made by the model out of the total samples. The accuracy assessment of Thorax disease classification methods is portrayed in Fig 6a. Here, the EnAE is key to extracting critical attributes using stacked auto-encoders combined with an attention module, EnAE extracts important features, removing noise and irrelevant data. Additionally, IMSTrans utilizes advanced attention mechanisms and hierarchical feature extraction,

**Table 2. Definition for Assessment Measures.**

| Measures | Formula | Symbol Definition |
|---|---|---|
| Accuracy | $\frac{TP+TN}{TP+TN+FP+FN}$ | *TP* True Positive<br>*TN* True Negative<br>*FP* False Positive<br>*FN* False Negative |
| Precision | $\frac{TP}{TP+FP}$ | |
| Recall | $\frac{TP}{TP+FN}$ | |
| F-Measure | $2 \times \frac{Precision \times Recall}{Precision+Recall}$ | |
| MCC | $MCC = \frac{TP \cdot TN - FP \cdot FN}{\sqrt{(TP+FP)(TP+FN)(TN+FP)(TN+FN)}}$ | |
| MAE | $MAE = \frac{1}{n}\sum_{i=1}^{n}\left|y_i - \hat{y}_i\right|$ | $\hat{y}_i$ -predicted value<br>$y_i$ -actual value<br>*n* -number of observation |

enabling it to process image patches efficiently. With values ranging from 0.878 at 50% training to 0.964 at 90%, the suggested model achieves the highest accuracy and routinely outperforms alternative approaches. Thus, the proposed EnAE + ChWO+IMSTrans model captures both local and global patterns in the X-rays, which leads to more accurate predictions compared to existing methods. The precision analysis is illustrated in Fig 6b. Here, the ChWO algorithm is utilized to select the most significant features after extraction. By discarding less relevant attributes, the model is streamlined to focus on features that are strongly correlated with specific thoracic diseases, reducing the risk of making incorrect classifications. The model's highly accurate performance in predicting positives is demonstrated by the proposed method's

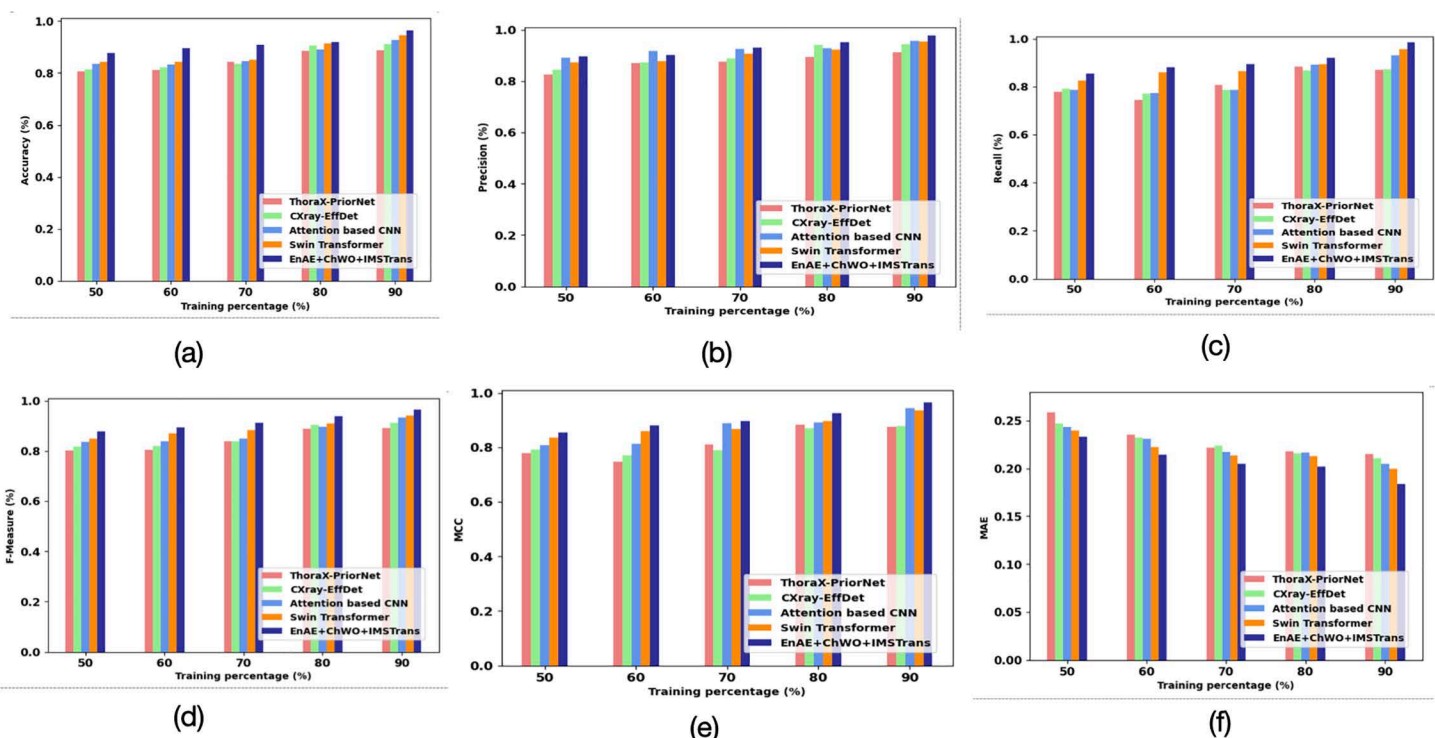

**Fig 6. Assessment Using Dataset 1: (a) Accuracy, (b) Precision, (c) Recall, (d) F-Measure, (e) MCC and (f) MAE.**

highest precision of 0.977 at 90% training percentage. Thus, the incorrect classifications are eliminated by the proposed model, which leads to higher precision by the proposed EnAE + ChWO+IMSTrans model. The recall-based analysis is illustrated in Fig 6c. The proposed method achieves higher recall due to the balanced data and efficient feature extraction modules. By augmenting the minority class samples, the model becomes better at identifying actual cases of Thorax diseases, which helps improve its recall. In addition, the EnAE captures finer and more complex features, ensuring that the model can detect even minor indicators of thoracic diseases, reducing false negatives. With a recall of 0.9845 at a 90% training percentage, the suggested technique accurately identifies the majority of relevant instances.

The F1-score-based analysis is demonstrated in Fig 6d The EnAE's ability to extract significant features and IMSTrans's robust classification ensure that both precision and recall are maximized. It indicates that the proposed model helps accomplish both sensitive in detecting diseases (high recall) and specificity in making correct predictions (high precision). The maximum F1-Score of 0.964 is obtained by the proposed method at 90% training percentage, demonstrating an effective trade-off between recall and precision. The Matthews Correlation Coefficient (MCC) is used to analyze the robustness of the Thorax disease model for both positive and negative classes. The MCC-based analysis is demonstrated in Fig 6e. The use of data augmentation helps reduce the bias toward majority classes that results in a more balanced classification. It leads to lower false positive and false negative rates, which contribute to a higher MCC score by the proposed EnAE + ChWO+IMSTrans model. The proposed method achieves the maximum MCC of 0.9647 at 90%, indicating that it is the most dependable and balanced model. The MAE-based analysis is illustrated in Fig 6f. EnAE minimizes reconstruction error when extracting features, which ensures that the latent representations of the X-rays are highly accurate. It reduces the overall error in classifying the Thorax disease. The careful selection of features helps minimize prediction errors. By ensuring that only relevant features are used, the model avoids making large errors, which keeps the MAE low. The proposed method has the lowest MAE of 0.184 at 90%, meaning it predicts with the least amount of error. Table 3 shows the Assessment Using Dataset 1

**4.2.2. Assessment using dataset 2.** The assessment of Thorax disease classification method using the D2 dataset is evaluated based on training data and is elaborated in this section.

Fig 7 analyzes the performance of various Thorax disease classification techniques on the D2 dataset, comparing their efficacy at different training data proportions, ranging from 50% to 90%. The accuracy is shown in Fig 7a, where EnaE+CHWO+IMSTrans continuously attains the best accuracy, hitting roughly 0.96 at 90% training data. Likewise, in Fig 7b Precision, EnaE+CHWO+IMSTrans is in the lead with approximately 0.95 at 90%. In Fig 7c, the Recall exhibits a comparable pattern, with EnaE+CHWO+IMSTrans reaching roughly 0.94. The F-Measure in Fig 7d, which strikes a balance between recall and precision, likewise peaks for EnaE+CHWO+IMSTrans at about 0.94. Fig 7e shows that the MCC scores for all models generally improve as the training percentage rises from 50% to 90%. In particular, proposed EnAe+CHWO+IMSTrans attains the greatest MCC with 90% training data, at roughly 0.96. At the maximum training, ThoraX-PriorNet has the lowest MCC, which is approximately 0.86. This suggests that, particularly when trained on more data, EnAe+CHWO+IMSTrans exhibits the best overall classification performance. Figure (f) shows that the MAE for all models tends to decrease as the training set size increases. EnAe+CHWO+IMSTrans shows the lowest MAE at the 90% training level, which is at 0.04, indicating the most minor average magnitude of errors in its predictions. At 90% training data, ThoraX-PriorNet has the greatest MAE, approximately 0.09, indicating higher average prediction errors. This suggests that, when trained with a larger dataset, EnAe+CHWO+IMSTrans produces the most accurate numerical predictions among all the models compared. Table 4 shows the assessment using Dataset 2.

**4.3. Accuracy-Loss:** Accuracy-loss analysis compares the model's accuracy and loss during both training and testing phases to assess the performance of the model and determine if it's generalizing well to unseen data.

The accuracy and loss trends for Dataset 1 to classify thorax disease over 100 epochs are displayed in Fig 8. Over 100 epochs, the accuracy trends are shown in Fig 8a. The accuracy of training rises quickly, peaking at about 95% around epoch 30 before plateauing. The trajectory of the testing accuracy is similar, with modest swings after a slightly lower peak

**Table 3. Assessment Using Dataset 1.**

| Methods/ Training Percentage | 50 | 60 | 70 | 80 | 90 |
|---|---|---|---|---|---|
| Accuracy | | | | | |
| ThoraX-PriorNet | 0.806 | 0.812 | 0.843 | 0.884 | 0.887 |
| CXray-EffDet | 0.813 | 0.822 | 0.835 | 0.905 | 0.912 |
| Attention based CNN | 0.836 | 0.834 | 0.846 | 0.891 | 0.928 |
| Swin Transformer | 0.842 | 0.844 | 0.851 | 0.913 | 0.945 |
| **Proposed** | **0.878** | **0.896** | **0.909** | **0.92** | **0.964** |
| **Precision** | | | | | |
| ThoraX-PriorNet | 0.8265 | 0.8705 | 0.876 | 0.8942 | 0.9115 |
| CXray-EffDet | 0.8452 | 0.872 | 0.8885 | 0.94 | 0.944 |
| Attention based CNN | 0.891 | 0.917 | 0.9255 | 0.929 | 0.958 |
| Swin Transformer | 0.8737 | 0.8794 | 0.9078 | 0.9239 | 0.9534 |
| **Proposed** | **0.896** | **0.903** | **0.931** | **0.9515** | **0.977** |
| **Recall** | | | | | |
| ThoraX-PriorNet | 0.778 | 0.7455 | 0.806 | 0.8825 | 0.8715 |
| CXray-EffDet | 0.7925 | 0.77 | 0.7865 | 0.867 | 0.874 |
| Attention based CNN | 0.7855 | 0.772 | 0.785 | 0.8915 | 0.931 |
| Swin Transformer | 0.8256 | 0.8597 | 0.8662 | 0.8926 | 0.9578 |
| **Proposed** | **0.855** | **0.8815** | **0.8935** | **0.921** | **0.9845** |
| **F1-Score** | | | | | |
| ThoraX-PriorNet | 0.8023 | 0.8041 | 0.8392 | 0.8885 | 0.892 |
| CXray-EffDet | 0.8178 | 0.8195 | 0.838 | 0.9045 | 0.9118 |
| Attention based CNN | 0.8367 | 0.8385 | 0.8502 | 0.8955 | 0.9337 |
| Swin Transformer | 0.8481 | 0.8712 | 0.8842 | 0.9089 | 0.9423 |
| **Proposed** | **0.8768** | **0.894** | **0.9125** | **0.9385** | **0.964** |
| **MCC** | | | | | |
| ThoraX-PriorNet | 0.7793 | 0.7476 | 0.8102 | 0.8827 | 0.8743 |
| CXray-EffDet | 0.7924 | 0.7712 | 0.7888 | 0.8697 | 0.8771 |
| Attention based CNN | 0.8074 | 0.8128 | 0.8887 | 0.8914 | 0.9428 |
| Swin Transformer | 0.8355 | 0.8605 | 0.8686 | 0.8976 | 0.9368 |
| **Proposed** | **0.8558** | **0.8819** | **0.8971** | **0.9242** | **0.9647** |
| **MAE** | | | | | |
| ThoraX-PriorNet | 0.2585 | 0.2354 | 0.2218 | 0.2182 | 0.2148 |
| CXray-EffDet | 0.2473 | 0.2326 | 0.2237 | 0.2161 | 0.2106 |
| Attention based CNN | 0.2434 | 0.231 | 0.2169 | 0.2164 | 0.2049 |
| Swin Transformer | 0.2401 | 0.2226 | 0.2137 | 0.2129 | 0.2001 |
| **Proposed** | **0.233** | **0.2143** | **0.2048** | **0.2017** | **0.184** |

at 94%. This shows that the model is capable of learning and generalizing to new data with reasonable accuracy. In Fig 8b, the loss curves are shown. Over time, the training loss decreases from roughly 0.7 to less than 0.2. After a brief period of volatility, the testing loss likewise drops dramatically, stabilizing at about 0.2. It is clear from the rather narrow difference between training and testing loss that the model is not significantly overfitting the training set.

The accuracy and loss trends for Dataset 2 to classify thorax disease over 100 epochs are displayed in Fig 9. Minor overfitting is suggested by the testing loss, which drops to a minimum of 0.21 before slightly increasing to 0.22, while the

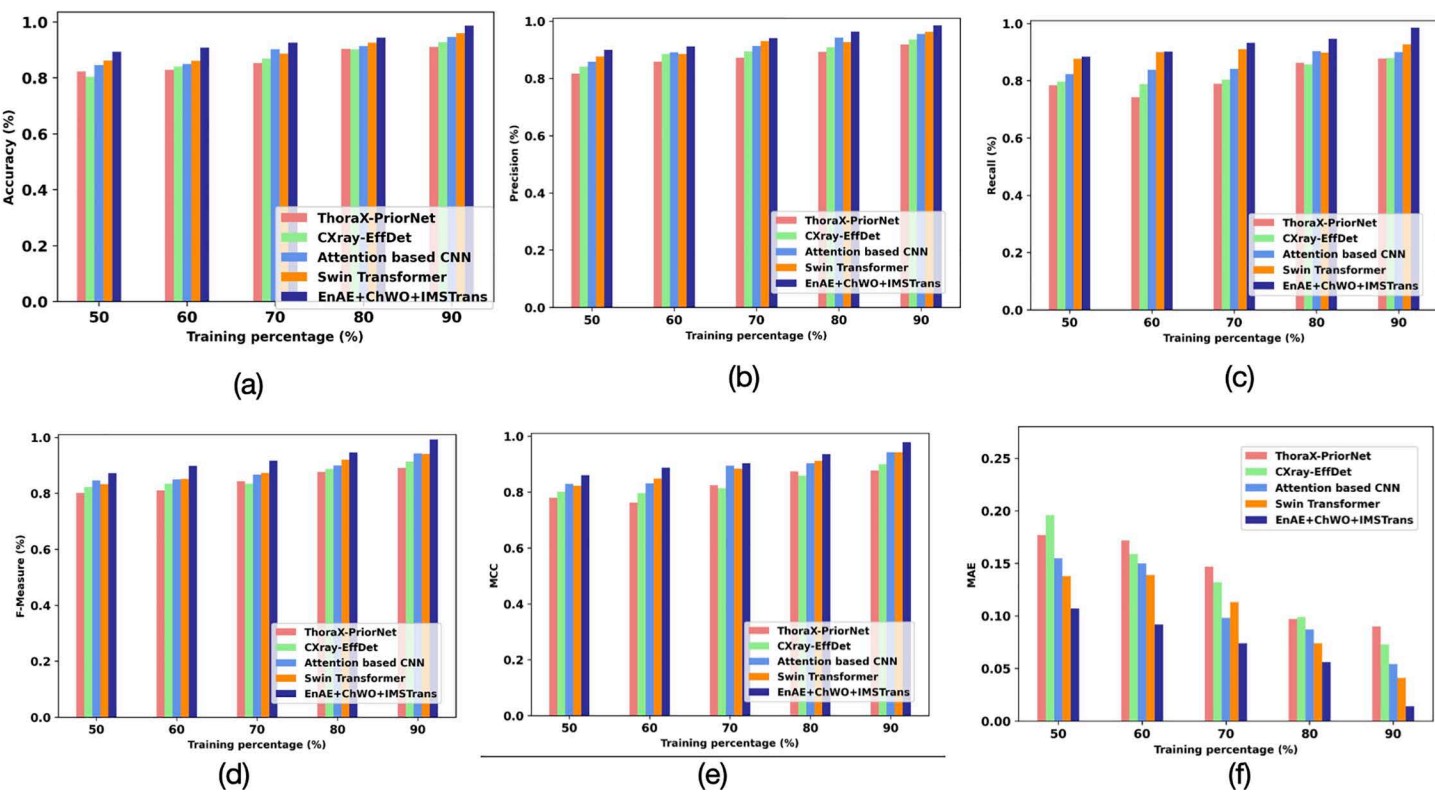

**Fig 7. Assessment Using Dataset 2: (a) Accuracy, (b) Precision, (c) Recall, (d) F-Measure, (e) MCC and (f) MAE.**

training loss gradually reduces from roughly 0.7 to 0.19. On the other hand, the black training accuracy quickly increases from 0.5 to a plateau at 0.97. Also improving is the testing accuracy, which peaks at about 0.95 and stays steady. Although there is some overfitting in later epochs, the model's overall high testing accuracy of almost 95% shows that learning is effective.

**4.4. Confusion Matrix:** The confusion matrix representation of the proposed EnAE+ChWO+IMSTrans model using dataset 1 and dataset 2 is portrayed in Fig 10, which illustrates the correct classification made by the proposed method.

Fig 10a shows a 14x14 matrix for thorax disease classification. For several classes, including class 0 (82), class 2 (71), and class 4 (107), the diagonal displays high correct classification rates. Misclassifications sometimes occur, though; for example, 37 cases of class 1 were incorrectly classified as class 0. The three categories of fibrosis, normal, and pneumonia are simplified in Fig 10b. With 250 pneumonia cases correctly predicted, 201 normal cases correctly classified, and 573 fibrosis patients correctly recognized, the matrix shows strong results. Three cases of pneumonia were misclassified as fibrosis, while two cases of normal were misclassified as fibrosis. At 0.986, this classification's overall accuracy score on Dataset 2 is remarkably high. The model's accuracy in classifying thoracic disorders is clearly demonstrated by these matrices, with Dataset 2 exhibiting robust findings.

**4.5. AUC Analysis:** The AUC analysis of Dataset 1 is portrayed in Fig 11a, b. The area under the ROC curve ranges from 0 to 1. A higher AUC value by EnAE+ChWO+IMSTrans model indicates better performance in distinguishing between classes. The analysis of Dataset 2's Area Under the Curve (AUC) is shown in Fig 12a, b. Various models' performances are evaluated in Fig 12a, and the model with the most excellent AUC, EnAE+ClWO+IMSTrans, shows superior

**Table 4. Assessment Using Dataset 2.**

| Methods/ Training Percentage | 50 | 60 | 70 | 80 | 90 |
|---|---|---|---|---|---|
| Accuracy | | | | | |
| ThoraX-PriorNet | 0.823 | 0.828 | 0.853 | 0.903 | 0.91 |
| CXray-EffDet | 0.804 | 0.841 | 0.868 | 0.901 | 0.927 |
| Attention based CNN | 0.845 | 0.85 | 0.902 | 0.913 | 0.946 |
| Swin Transformer | 0.862 | 0.861 | 0.887 | 0.926 | 0.959 |
| **Proposed** | **0.893** | **0.908** | **0.926** | **0.944** | **0.986** |
| Precision | | | | | |
| ThoraX-PriorNet | 0.817 | 0.859 | 0.872 | 0.892 | 0.918 |
| CXray-EffDet | 0.841 | 0.885 | 0.894 | 0.909 | 0.936 |
| Attention based CNN | 0.858 | 0.891 | 0.913 | 0.942 | 0.955 |
| Swin Transformer | 0.876 | 0.885 | 0.931 | 0.926 | 0.962 |
| **Proposed** | **0.899** | **0.911** | **0.941** | **0.963** | **0.985** |
| Recall | | | | | |
| ThoraX-PriorNet | 0.784 | 0.742 | 0.789 | 0.862 | 0.877 |
| CXray-EffDet | 0.797 | 0.788 | 0.804 | 0.856 | 0.879 |
| Attention based CNN | 0.823 | 0.838 | 0.841 | 0.903 | 0.9 |
| Swin Transformer | 0.876 | 0.899 | 0.91 | 0.897 | 0.926 |
| **Proposed** | **0.883** | **0.902** | **0.932** | **0.946** | **0.985** |
| F1-Score | | | | | |
| ThoraX-PriorNet | 0.801 | 0.81 | 0.843 | 0.876 | 0.89 |
| CXray-EffDet | 0.822 | 0.834 | 0.834 | 0.887 | 0.913 |
| Attention based CNN | 0.846 | 0.849 | 0.867 | 0.9 | 0.942 |
| Swin Transformer | 0.833 | 0.851 | 0.873 | 0.92 | 0.94 |
| **Proposed** | **0.872** | **0.897** | **0.917** | **0.946** | **0.993** |
| MCC | | | | | |
| ThoraX-PriorNet | 0.779 | 0.762 | 0.825 | 0.874 | 0.877 |
| CXray-EffDet | 0.801 | 0.797 | 0.814 | 0.859 | 0.9 |
| Attention based CNN | 0.829 | 0.831 | 0.894 | 0.903 | 0.942 |
| Swin Transformer | 0.823 | 0.848 | 0.883 | 0.911 | 0.942 |
| **Proposed** | **0.861** | **0.887** | **0.903** | **0.935** | **0.978** |
| MAE | | | | | |
| ThoraX-PriorNet | 0.177 | 0.172 | 0.147 | 0.097 | 0.09 |
| CXray-EffDet | 0.196 | 0.159 | 0.132 | 0.099 | 0.073 |
| Attention based CNN | 0.155 | 0.15 | 0.098 | 0.087 | 0.054 |
| Swin Transformer | 0.138 | 0.139 | 0.113 | 0.074 | 0.041 |
| **Proposed** | **0.107** | **0.092** | **0.074** | **0.056** | **0.014** |

classification abilities. The AUC curves for several classes (0, 1, and 2) are shown in Fig 12b. All of these classes achieve unusually high AUC scores of 0.99, indicating superior discrimination ability across all categories of Dataset 2.

**4.6. Convergence Analysis:** Convergence analysis evaluates how quickly and effectively an optimization algorithm minimizes the loss function, ensuring that the model reaches the global or near-global minimum efficiently. The convergence analysis is demonstrated in Fig 13. The inclusion of chaotic mapping assist the proposed ChWO algorithm to converge faster compared to existing whale optimization algorithm.

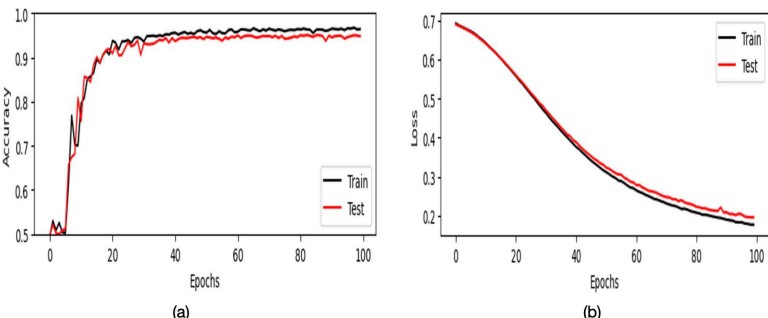

**Fig 8. Accuracy-Loss Analysis for Dataset 1.**

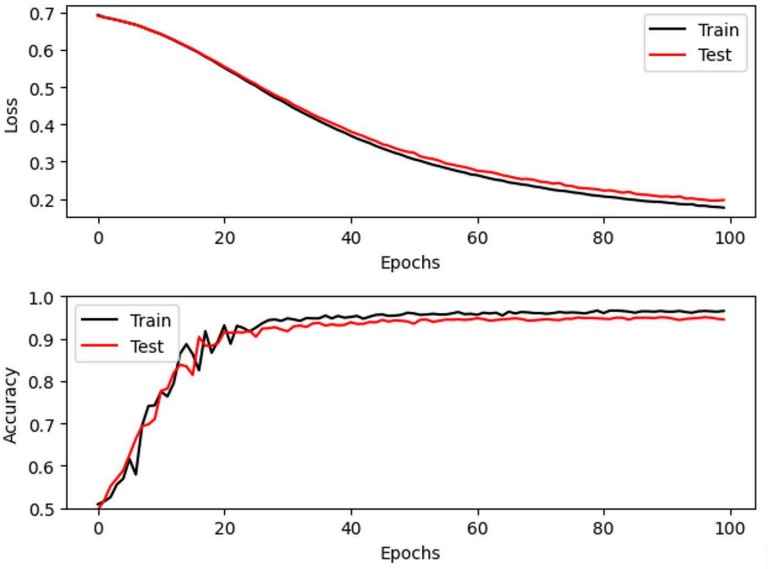

**Fig 9. Accuracy-Loss Analysis for Dataset 2.**

## 4.7. Comparative discussion

The comparative discussion for Dataset 1 and Dataset 2 based on the best case with 90% of training data and 10% of testing data is portrayed in Table 5. The proposed EnAE+ChWO+IMSTrans approach performs better across all assessment measures, according to a comparative analysis of Thorax disease categorization on Datasets 1 and 2. The proposed approach outperformed ThoraX-PriorNet, CXray-EffDet, Attention-based CNN, and Swin Transformer by 7.99%, 5.39%, 3.73%, and 1.97%, respectively, and obtained the highest accuracy (0.964) for Dataset 1. Likewise, the proposed technique outperformed Swin Transformer, the following best method, by 2.81% for Dataset 2, an accuracy of 0.986. Throughout both datasets, the proposed approach continuously yielded superior outcomes in terms of accuracy, recall, F-score, and MCC. Its robustness in detecting true positives is noteworthy, as evidenced by the recall on Datasets 1 (0.9845) and 2 (0.985). Additionally, the model's Mean Absolute Error (MAE) was the lowest, particularly on Dataset 2, at 0.014, which is far lower than all baseline techniques. These findings highlight the improved generality and efficacy of the proposed method in classifying Thorax disease, particularly when 90% of the data is used for training and 10% is used for testing.

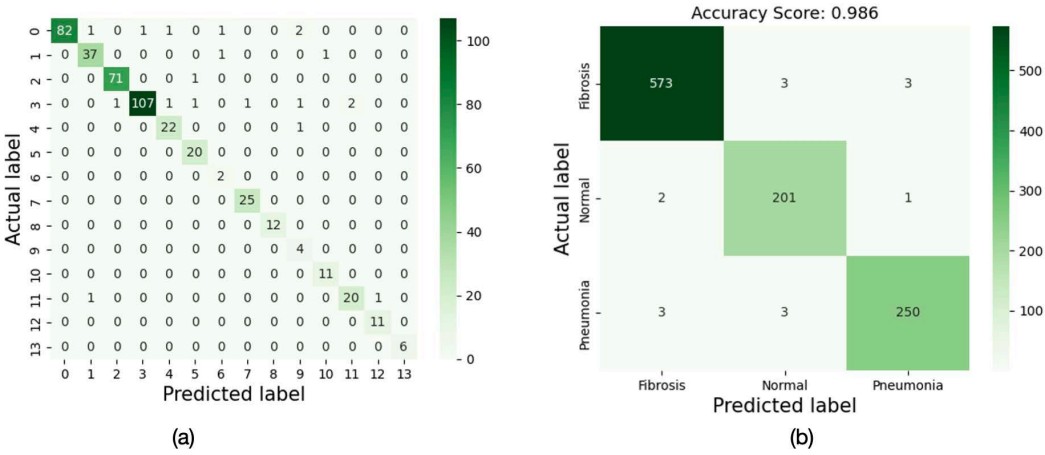

**Fig 10. Confusion Matrix for (a) Dataset 1 and (b) Dataset 2.**

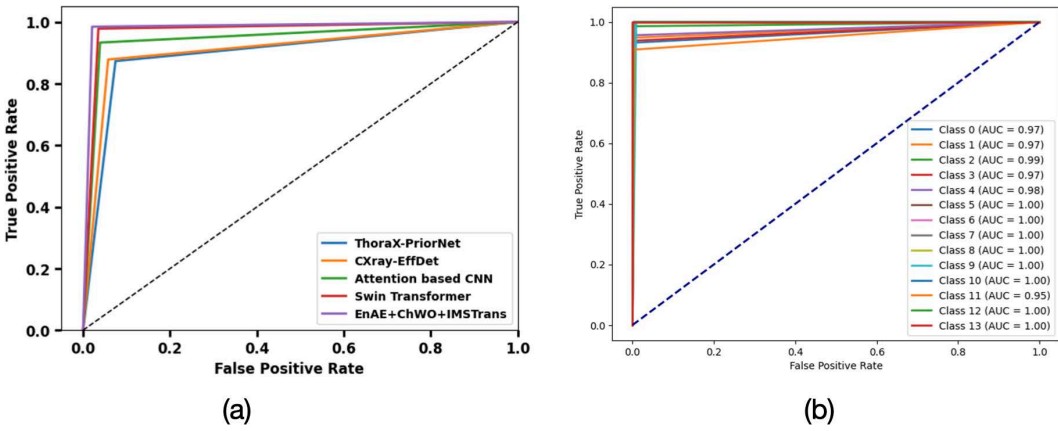

**Fig 11. AUC Analysis for Dataset 1.**

Overall, the consistent superiority of the two datasets highlights their suitability for dependable clinical use in medical image classification tasks.

Here, the comparative analysis portrays the superiority of the proposed model compared to the existing methods.

A comparative analysis of proposed and state-of-the-art methods for classifying thoracic diseases across two datasets is shown in Table 6. The suggested approach achieves the highest accuracy, precision, and recall at 90%, outperforming all others. It considerably outperforms CycleGAN with 93.3% accuracy, 94.7% precision, and 92.1% recall on Dataset 1, achieving 96.4% accuracy, 97.7% precision, and 98.45% recall. Likewise, the suggested approach attains 98.6% accuracy, 98.5% precision, and 98.5% recall in Dataset 2. On the other hand, DenseNet121 and CX-Ultranet perform moderately, demonstrating the superior efficacy of the proposed model in classifying thoracic diseases.

## 5. Conclusion

The proposed approach incorporates several key innovations, including data augmentation during pre-processing to mitigate class imbalance, feature extraction using the EnAE with an attention mechanism and optimal feature selection

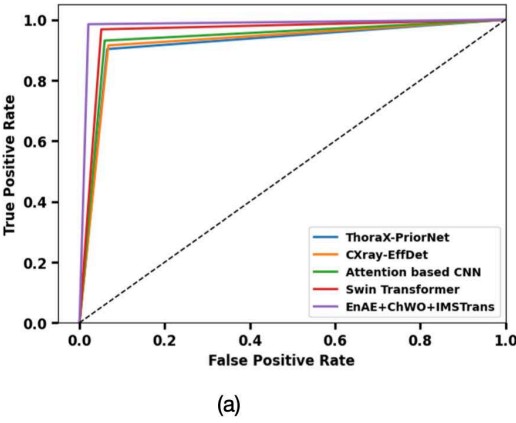
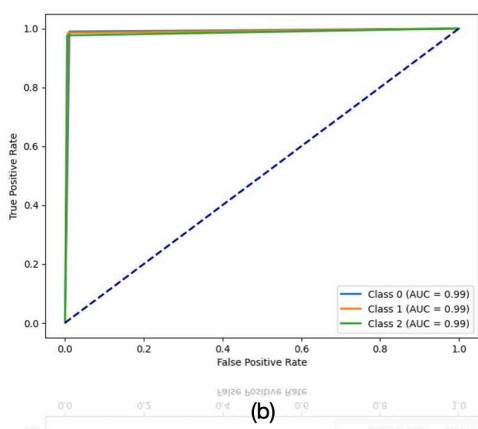

(a)    (b)

**Fig 12. AUC Analysis for Dataset 2.**

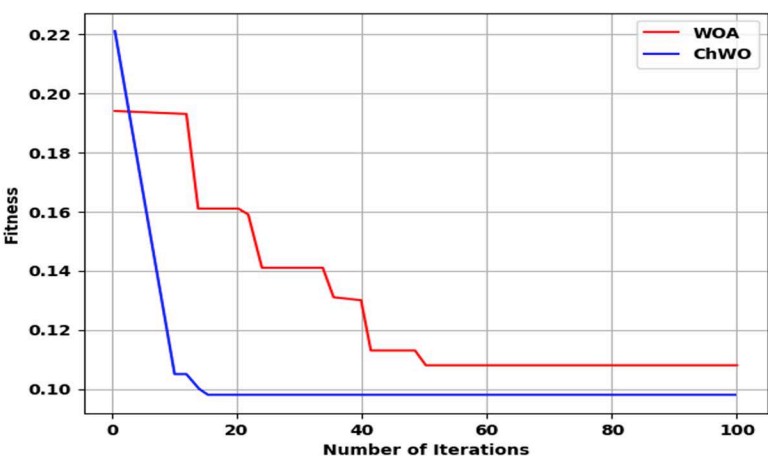

**Fig 13. Convergence Analysis.**

through the ChWO algorithm. Finally, classification is performed using the IMSTrans model, which has demonstrated superior performance in terms of accuracy, precision, recall, and overall classification efficacy compared to existing methods. The proposed model was rigorously validated using two major datasets: the Chest X-Ray dataset with over 112,000 images and the Lung Disease Dataset of 4,260 high-resolution images. Results showed superior performance, achieving accuracies of up to 96.4% and 98.6% on these datasets, respectively. The proposed framework is computationally intensive, particularly during the training phase, requiring significant resources for both training and feature extraction.

### 5.1. Limitations and future work

The proposed method faces certain limitations, despite showing improved classification performance when employing the Chaotic Whale Optimization Algorithm for feature selection and the EnAE with attention mechanisms. First and foremost, it is computationally demanding, particularly during the training and feature extraction stages, using significant resources that could impede its use in environments with limited resources, such as remote hospitals or mobile diagnostic units. Furthermore, the quality and representativeness of the Chest X-ray dataset used are crucial to the model's performance;

**Table 5. Comparative Analysis for Datasets 1 and 2.**

| Methods/ Metrics | Accuracy | Precision | Recall | F-Score | MCC | MAE |
|---|---|---|---|---|---|---|
| Dataset 1 | | | | | | |
| ThoraX-PriorNet | 0.887 | 0.9115 | 0.8715 | 0.892 | 0.8743 | 0.2148 |
| CXray-EffDet | 0.912 | 0.944 | 0.874 | 0.9118 | 0.8771 | 0.2106 |
| Attention based CNN | 0.928 | 0.958 | 0.931 | 0.9337 | 0.9428 | 0.2049 |
| Swin Transformer | 0.945 | 0.9534 | 0.9578 | 0.9423 | 0.9368 | 0.2001 |
| **Proposed** | **0.964** | **0.977** | **0.9845** | **0.964** | **0.9647** | **0.184** |
| Dataset 2 | | | | | | |
| ThoraX-PriorNet | 0.91 | 0.918 | 0.877 | 0.89 | 0.877 | 0.09 |
| CXray-EffDet | 0.927 | 0.936 | 0.879 | 0.913 | 0.9 | 0.073 |
| Attention based CNN | 0.946 | 0.955 | 0.9 | 0.942 | 0.942 | 0.054 |
| Swin Transformer | 0.959 | 0.962 | 0.926 | 0.94 | 0.942 | 0.041 |
| **Proposed** | **0.986** | **0.985** | **0.985** | **0.993** | **0.978** | **0.014** |

**Table 6. Comparative Analysis of proposed and state-of-the-art methods.**

| Dataset 1 | | | |
|---|---|---|---|
| Methods/ Metrics | Accuracy | Precision | Recall |
| CX-Ultranet [30] | 0.844 | 0.857 | 0.835 |
| DenseNet121 [31] | 0.861 | 0.874 | 0.849 |
| CycleGAN [32] | 0.933 | 0.947 | 0.921 |
| **Proposed (90%)** | **0.964** | **0.977** | **0.9845** |
| **Dataset 2** | | | |
| CX-Ultranet | 0.842 | 0.832 | 0.819 |
| DenseNet121 | 0.867 | 0.869 | 0.842 |
| CycleGAN | 0.949 | 0.956 | 0.942 |
| **Proposed (90%)** | **0.986** | **0.985** | **0.985** |

even with data augmentation efforts, imbalances and rare illness classes persist as problems. There is currently no test for generalizability to other medical imaging modalities, which restricts wider clinical use.

Thus, Future research should conduct comprehensive clinical validation tests with radiologists to overcome the uncertainties surrounding the model's applicability in actual clinical situations. To assess the model's interpretability, dependability, and practical utility in routine diagnostics, such investigations are crucial. Refinements can be guided by collaborations with doctors to better address clinical demands. Trust and usability will be improved by incorporating explainability elements. Furthermore, deployment requires optimizing computing efficiency, particularly in environments with restricted resources. It will be easier to move from a research prototype to a clinically useful tool for diagnosing thorax disease if the dataset is expanded to include multi-institutional and multi-modal imaging data.

## Author contributions

**Conceptualization:** Nadim Rana, Yahaya Coulibaly.

**Data curation:** Md Imran Alam, Zeba Khan, Mohammad Zubair Khan.

**Formal analysis:** Nadim Rana, Yahaya Coulibaly.

**Funding acquisition:** Yahaya Coulibaly.

**Investigation:** Ayman Noor.

**Methodology:** Nadim Rana.

**Project administration:** Mohammad Zubair Khan.

**Software:** Md Imran Alam, Zeba Khan.

**Supervision:** Yahaya Coulibaly, Mohammad Zubair Khan.

**Validation:** Ali Tahir.

**Visualization:** Talal H. Noor.

**Writing – original draft:** Nadim Rana, Yahaya Coulibaly.

**Writing – review & editing:** Yahaya Coulibaly.

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
