## [Decision Letter · Decision Letter 0]

Dear Dr. Coulibaly,

Thank you for submitting your manuscript to PLOS ONE. After careful consideration, we feel that it has merit but does not fully meet PLOS ONE’s publication criteria as it currently stands. Therefore, we invite you to submit a revised version of the manuscript that addresses the points raised during the review process.

We look forward to receiving your revised manuscript.

Kind regards,

Mohammad Alfrad Nobel Bhuiyan, Ph.D.

Academic Editor

PLOS ONE

**Journal Requirements:**

1. When submitting your revision, we need you to address these additional requirements. Please ensure that your manuscript meets PLOS ONE's style requirements, including those for file naming. The PLOS ONE style templates can be found at https://journals.plos.org/plosone/s/file?id=wjVg/PLOSOne_formatting_sample_main_body.pdf and https://journals.plos.org/plosone/s/file?id=ba62/PLOSOne_formatting_sample_title_authors_affiliations.pdf 2. Please note that PLOS ONE has specific guidelines on code sharing for submissions in which author-generated code underpins the findings in the manuscript. In these cases, we expect all author-generated code to be made available without restrictions upon publication of the work. Please review our guidelines at https://journals.plos.org/plosone/s/materials-and-software-sharing#loc-sharing-code and ensure that your code is shared in a way that follows best practice and facilitates reproducibility and reuse. 3. We note that your Data Availability Statement is currently as follows: All relevant data are within the manuscript and its Supporting Information files. Please confirm at this time whether or not your submission contains all raw data required to replicate the results of your study. Authors must share the “minimal data set” for their submission. PLOS defines the minimal data set to consist of the data required to replicate all study findings reported in the article, as well as related metadata and methods (https://journals.plos.org/plosone/s/data-availability#loc-minimal-data-set-definition). For example, authors should submit the following data: - The values behind the means, standard deviations and other measures reported;- The values used to build graphs;- The points extracted from images for analysis. Authors do not need to submit their entire data set if only a portion of the data was used in the reported study. If your submission does not contain these data, please either upload them as Supporting Information files or deposit them to a stable, public repository and provide us with the relevant URLs, DOIs, or accession numbers. For a list of recommended repositories, please see https://journals.plos.org/plosone/s/recommended-repositories. If there are ethical or legal restrictions on sharing a de-identified data set, please explain them in detail (e.g., data contain potentially sensitive information, data are owned by a third-party organization, etc.) and who has imposed them (e.g., an ethics committee). Please also provide contact information for a data access committee, ethics committee, or other institutional body to which data requests may be sent. If data are owned by a third party, please indicate how others may request data access. 4. PLOS requires an ORCID iD for the corresponding author in Editorial Manager on papers submitted after December 6th, 2016. Please ensure that you have an ORCID iD and that it is validated in Editorial Manager. To do this, go to ‘Update my Information’ (in the upper left-hand corner of the main menu), and click on the Fetch/Validate link next to the ORCID field. This will take you to the ORCID site and allow you to create a new iD or authenticate a pre-existing iD in Editorial Manager.

Reviewers' comments:

Reviewer's Responses to Questions

**Comments to the Author**

1. Is the manuscript technically sound, and do the data support the conclusions?

Reviewer #1: Yes

Reviewer #2: Yes

Reviewer #3: Yes

2. Has the statistical analysis been performed appropriately and rigorously?

Reviewer #1: N/A

Reviewer #2: Yes

Reviewer #3: I Don't Know

3. Have the authors made all data underlying the findings in their manuscript fully available?

Reviewer #1: No

Reviewer #2: Yes

Reviewer #3: Yes

4. Is the manuscript presented in an intelligible fashion and written in standard English?

Reviewer #1: Yes

Reviewer #2: Yes

Reviewer #3: Yes

**Reviewer #1:**  Review of the Paper: "Improved Swin Transformer based Thorax Disease Classification with Optimal Feature Selection using Chest X-ray"

The paper proposes a novel method for classifying thoracic diseases using a deep learning framework that combines an Improved Swin Transformer for classification, Chaotic Whale Optimization for feature selection, and Enhanced Auto-Encoder for feature extraction. However, there are some restrictions need to be solved.

1. There is uncertainty over the model's applicability in real-world clinical operations. There hasn't been any clinical validation with radiologists.

2. The proposed method only compares with limited number of state-of-the-art methods and also ignores newer methods.

3. There are different optimization algorithms that can be use instead of Chaotic Whale Optimization Algorithm. In the paper, do not mention any reason for using this algorithm. Additionally, ChWO algorithm adds extra computational overhead.

4. There are different number of chest x-ray datasets that can be used for evaluating the approach. Using just one dataset is not enough.

5. The dataset size and detailed information about it are not fully discussed.

6. The best results in the tables do not bold.

7. Overall, the article is not well-published and well-structured, especially in the Result and Discussion section.

**Reviewer #2: ** Strengths of the Paper:

1. The proposed framework effectively addresses class imbalance through data augmentation, enhancing model generalization.

2. The combination of stacked auto-encoder architecture with an attention module in EnAE improves feature representation and classification accuracy.

3. The utilization of the Chaotic Whale Optimization Algorithm refines feature selection, ensuring optimal attributes are used for classification.

4. The IMSTrans model efficiently processes high-dimensional medical image data, leading to improved classification performance.

5. The experimental results indicate the proposed method achieves high performance metrics, demonstrating its reliability and efficiency in thorax disease classification.

Main Comments for Improvement:

1. Recent References: The paper should include references from 2024, ensuring the literature review is up-to-date and reflects the latest advancements in thoracic disease classification and deep learning applications.

2. Comparison with State-of-the-Art: The results should be compared with recent state-of-the-art methods. Additionally, citations should be included in the comparative analysis to support claims of superior performance.

3. Acronyms Usage: The full form of each acronym should be provided only at its first mention in the text. Subsequent mentions should use only the acronym to maintain readability and consistency.

4. Problem Statement References: The problem statement section should be supported with appropriate references to establish the significance and relevance of the study.

5. Proposed Methodology References: The methodology should be reinforced with citations, particularly when discussing techniques such as EnAE, the Chaotic Whale Optimization Algorithm, SWIN, and IMSTrans, to establish credibility and provide background for the employed techniques.

6. Data Distribution Analysis: The paper does not mention the data distributions before and after augmentation. It is essential to provide details on the original dataset distribution and how augmentation affects class balance and feature diversity.

7. Method Limitations and Future Work: The paper should discuss the limitations of the proposed method, such as computational complexity, dependency on dataset quality, or generalizability to other medical imaging modalities. Additionally, future research directions should be outlined.

**Reviewer #3: ** Thank you for the opportunity to review " Improved Swin Transformer based Thorax Disease Classification with Optimal Feature Selection using Chest X-ray”.

While the topic of the submitted article of great importance, minor concerns exist regarding the current version of the manuscript.

The introduction needs a clearer articulation of the research gap and objectives.

The literature review should have more critical analysis of prior work. Rather than merely summarizing existing research, the authors should highlight how their study builds upon and differentiates from previous efforts.

The manuscript mentions a publicly available dataset but lacks specifics (e.g., class distribution, preprocessing steps like handling missing or noisy data). Clarify these details to ensure reproducibility.

Provide more details on hyperparameter tuning (e.g., learning rates, batch sizes) and any cross-validation strategies used.

In discussion section, consider including a comparison of findings with prior studies to highlight the novelty and significance of the work.

Some citations (e.g., [21]–[32]) are formatted inconsistently. Ensure all references follow PLOS ONE guidelines.

• "ChextX-Ray dataset" → "ChestX-Ray dataset" (Line # 434).

• "publically" → "publicly" (Line # 191, 434).

**Do you want your identity to be public for this peer review?** For information about this choice, including consent withdrawal, please see our Privacy Policy

Reviewer #1: No

Reviewer #2: No

Reviewer #3: No

---

## [Author Response · Author response to Decision Letter 1]

8 Jun 2025

Dear reviewer and editor,

We have carefully revised the manuscript based on all the comments provided by the reviewers. Each point has been addressed in detail, and a separate Response to Reviewers document has been prepared accordingly. The revised manuscript now includes the following improvements:

• Clarification of the research gap and study objectives in the Introduction.

• Enhanced critical analysis in the literature review and proper citations for recent works (2024).

• Justification for the use of Chaotic Whale Optimization, and a discussion of alternative methods.

• Added dataset description, including class distribution, preprocessing steps, and augmentation effects.

• Inclusion of method limitations and suggestions for future research.

• Standardization of references and correction of typographical errors.

• Improved structure and clarity in the Results and Discussion section.

We have uploaded the following documents with this resubmission:

• A Response to Reviewers’ letter detailing how each comment was addressed.

• A Revised Manuscript with Track Changes, highlighting all modifications.

• A Clean Copy of the Revised Manuscript without track changes.

Kindly access to those files to see the required changes.

Regards,

---

## [Editor Report · Decision Letter 1]

Improved Swin Transformer based Thorax Disease Classification with Optimal Feature Selection using Chest X-ray

PONE-D-24-53299R1

Dear Dr. Coulibaly,

We’re pleased to inform you that your manuscript has been judged scientifically suitable for publication and will be formally accepted for publication once it meets all outstanding technical requirements.

Kind regards,

Mohammad Alfrad Nobel Bhuiyan, Ph.D.

Academic Editor

PLOS ONE
---

## [Editor Report · Acceptance letter]

PONE-D-24-53299R1

PLOS ONE

Dear Dr. Coulibaly,

I'm pleased to inform you that your manuscript has been deemed suitable for publication in PLOS ONE. Congratulations! Your manuscript is now being handed over to our production team.

Kind regards,

on behalf of

Assistant Professor Mohammad Alfrad Nobel Bhuiyan

Academic Editor

PLOS ONE